# Social Progress beyond GDP: A Principal Component Analysis (PCA) of GDP and Twelve Alternative Indicators

**Bing Wang and Tianchi Chen \*** 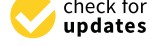

College of Public Administration, Huazhong University of Science and Technology (HUST), Wuhan 430074, China; wbyf@hust.edu.cn
\*   Correspondence: d202081314@hust.edu.cn

**Abstract:** What social progress is and how to measure it are seemingly plain but essentially intricate questions that have not been clarified to date, which has led to various social problems and development failures. Designed after the Great Depression in the 1930s, Gross Domestic Product (GDP) has been, on the one hand, regarded as the greatest invention of the 20th century and is widely accepted as the primary indicator for social progress, but on the other hand, it has been criticized as knowing the price of everything but the value of nothing. The Beyond GDP Movement that has been active since the 2010s has inspired global interest in designing indicators to replace or supplement GDP, but none of them stands out as GDP's successor. We take 12 influential indicators that consider beyond GDP and carry out a Principal Component Analysis (PCA) to investigate their correlations. The results indicate that GDP per capita (GDPP) can explain 65.61% of the information in the first principal component (PC) and account for 51.10% of the information related to the total 13 indicators, indicating its major role in social progress. Most indicators have strong correlations with GDPP, not beyond, and only the Ecological Footprint per capita (EFP) and Happy Planet Index (HPI) that have negative and weak correlations with GDPP, respectively, can provide new perspectives and values beyond GDP. Social progress is based upon various public values, and the indicators are the measurements of these values. Although GDP and economic values play major roles during social development, other indicators and their potential public values cannot be ignored. Prioritizing these public values and monitoring their indicators are essential to achieving sustainable and comprehensive social progress.

**Keywords:** beyond GDP; social progress; indicator; value; public value (PV); principal component analysis (PCA)

## 1. Introduction

The past several decades have witnessed continuously changing definitions and understandings of social progress with the advancement of the economy and technology and the transformation of values. Accordingly, the indicators that have been used for its measurement have constantly evolved. Gross Domestic Product (GDP), which was invented in the 1930–1940s after the Great Depression, is one of the most representative measures of economic well-being and social progress and has been regarded as the "greatest invention of the 20th century" [1,2]. However, it has also been severely criticized since its inception. One of its creators, Simon. S. Kuznets (1901–1985), clearly stated that a nation's welfare could scarcely be defined by its National Income or measured by GDP [3] (pp. 1–12). Briefly, GDP is mainly criticized for overlooking other social progress factors, such as subjective well-being, equity, and environmental sustainability [4].

The measure of GDP-based human development merely highlights economic well-being but ignores the harm caused by narrow economic growth, including inequality, low social satisfaction, and environmental degradation. According to the Genuine Progress Indicator (GPI) and other indicators for 17 countries occupying 53% of the global population,

global-scale economic welfare has not improved since 1978 despite prominent economic growth [5]. The costs of pursuing GDP growth may have gradually cancelled out the benefits that we have gained. In light of this, many scholars and practitioners have appealed to another development pattern that favors human well-being, social justice, and ecological sustainability, triggering influential academic endeavors such as the Social Indicators Movement in the 1960s [6], the Beyond GDP initiative by the European Commission in 2007 [4,7], and the United Nations Sustainable Development Goals (SDGs) in 2015 [8]. These endeavors have inspired an explosion of indicators. However, are they really beyond GDP?

The mainstream Beyond GDP indicators can be classified as a set of indicators and the compound index. The former usually refers to classifying indicators by topics or themes based on one or more levels without aggregation, such as the Measures of National Well-being Dashboard in the UK [9] and the Quality of Life Framework in Canada [10]. In contrast, the latter has been prevalent in recent decades, referring to merging data from a wide range of dimensions into a scalar value by assigning weights to the chosen variables [11]. Given that the aggregation of sub-indicators into a single index is easy to understand and monitor, it is favored by governments, academic institutions, and non-governmental organizations (NGOs).

Most of the indicators measuring social progress have shifted their focus from GDP to three other main aspects: human well-being, ecological sustainability, and a combination of the two. For instance, the Human Development Index (HDI) [12], which was created by the United Nations Development Programme (UNDP), measures human development in the dimensions of the economy, education, and healthcare. The Environmental Performance Index (EPI) [13], which was developed by the Yale Center for Environmental Law & Policy (YCELP), assesses the performance of ecological sustainability using two dimensions: environmental health and ecosystem vitality. The Happy Planet Index (HPI), which was designed by the New Economics Foundation (NEF) [14], evaluates "happiness" by combining both well-being and ecological sustainability.

Since all of these indicators are considered to be able to measure social progress in various dimensions, what are their merits and demerits, relationships, and reliability? Moreover, most research has focused on developing new indicators rather than on re-examining the correlations among the existing indicators in-depth to find potential problems and propose valid and updated theories on constructing more reasonable indicators. As a result, due to the lack of valid theories and insufficient and misguided indicators, social progress remains a process of trial-and-error experimentation with a high failure rate and uneven progress [15]. Particularly, numerous studies have indicated that the so-called Beyond GDP Movement is not as successful as anticipated, and the applications of those Beyond GDP indicators in national policymaking are facing enormous challenges [16–19]. Both developed and developing countries are encountering various difficulties in terms of social progress.

Therefore, to clarify the essence of GDP and other related indicators, this article chooses 12 influential indicators that are "beyond GDP" and then employs a Principal Component Analysis (PCA) to investigate their correlations with GDP per capita (GDPP). The purposes of this research are to explore whether and how these indicators are beyond GDP and if they can measure social progress. In Section 2, we argue that public value should be the basis of the indicators and the foundation of social progress. In Section 3, the data sources of the GDPP, the 12 alternative indicators, and the PCA as a methodology are explained. Section 4 discusses the PCA results. The interesting and enlightening findings are that (1) most of the studied indicators are not beyond GDP since they strongly correlate with GDPP and measure similar value dimensions; (2) only the Ecological Footprint per capita (EFP) that is strongly negatively correlated with GDPP and the Happy Planet Index (HPI) that has weak correlation with GDPP are enlightening and could be regarded as beyond GDP because they reveal opposing or different information comparing to the GDPP; and (3) despite the widespread critics, GDPP still plays a major role in the economic well-being and social development, as it can explain 65.61% of the information in the first principal

component (PC) and account for 51.10% of the information related to the total 13 indicators. Finally, some conclusions and recommendations are made.

## 2. Beyond GDP Movement, Public Value, and Indicators

### 2.1. Beyond GDP Movement

The Beyond GDP Movement was initially proposed at a conference organized by the European Commission (EC) in November 2007, with the aim of developing more inclusive indicators that are as clear and appealing as GDP. In 2009, the EC released its roadmap to improve the indicators that adjust, replace, and supplement GDP. Afterwards, a final document, "Progress on 'GDP and beyond' actions", also known as the Stiglitz–Sen–Fitoussi report, was released in the same year [7]. Since then, the Beyond GDP Movement has aroused the academic interests of not only the EU but also of global governments, triggering an explosion of indicators [20].

However, no alternative indicator stands out as a successor, and GDP remains dominant in economics, politics, and public policy for measuring social progress, constituting a "GDP Paradox" [21]. Although the recent 2020 GlobeScan-Ethical Markets Beyond GDP Survey found that an average of 72% per cent of the general public prefers expanding GDP with health, education, and environmental data, financial markets, economists, media, most governments, and companies are persistent with what is known as "GDP fetishism" [22]. The SDGs that have been advocated by the UN have also been criticized for their reliance on GDP, causing significant incoherence and the hinderance of the overall realization of policy goals [23]. Although some indicators have been applied in small-scale contexts, such as the Genuine Progress Indicator (GPI) adopted in Maryland [24], Beyond GDP efforts have so far been unsuccessful [18,19]. In Belgium, after the economic crisis of 2008–2012, alternative indicators were totally ignored in federal election campaigns, and boosting GDP growth has once again become the central task of the Belgian federal government. Meanwhile, Germany was the country favoring Beyond GDP indicators the most, but after 2013, it became the country that was the most likely to pursue long-term and continuous economic growth [16]. Canadian federal and provincial governments have not widely adopted Beyond GDP indicators, and thus the progress of the Beyond GDP Movement has been limited in this context as well [17].

### 2.2. Value and Public Value

Progress and development imply the potential to become better. Without a precise definition and understanding of 'betterness' and goodness, it is hard to declare progress and development. Value is the formal academic concept of goodness. As social psychologist Milton Rokeach [25] (p. 3) stated, "the value concept, more than any other, should occupy a central position ... able to unify the apparently diverse interests of all the sciences concerned with human behavior". Kluckhohn [26] defined value as "a conception, explicit or implicit, distinctive of an individual or characteristic of a group, of the desirable which influences the selection from available modes, means, and ends of action". Values exist in all social spheres and have profound impacts on our perception of reality, such as what is good or bad, and provide identity to individuals, groups, and organizations, as well as guide our behaviors [27]. Bozeman [28] argued that values are difficult to change but that it is possible to change them after careful deliberation. Over the past 50 years, western development theories have mainly been dominated by neo-liberal economics, which is based upon value monism and the human nature assumption of homo economicus, while real human society is based on value pluralism, comprising human nature of homo sociologicus and homo politicus [29–31]. Due to the incomparability and incommensurability of plural values, measuring social development by GDP and evaluating a wide range of social phenomena using monistic value is naive and misleading.

Public Value (PV) is inherited from the concept of value, meaning that value is collectively shared by the general public, such as wealth, equity, freedom, democracy, love, happiness, and technology [32]. Government can create or safeguard various public values

through services, laws, and regulations [33,34], so public value should be the principle of governance and policies. Benington [29] extended public value beyond the market and economy, identifying that ecology, politics, society, and culture all have their specific public values. Social institutions and norms are functional for segregating these spheres and protecting their values [35]. Social development should improve and enhance key public values in an orderly and synchronous way [36].

Although economic value is pivotal and fundamental, it ought not be the sole value. Public values share close and complex relationships in various dimensions. Economic value at the expense of other public values such as equity, environment, and health is common, but economic progress alone cannot be recognized as comprehensive and authentic social progress. In the background of unprecedented economic affluence but social and ecological crises, the governance of public value has been recognized as a new theoretical paradigm that is beyond traditional and new public management, which have been widely criticized for their failure regarding public value due to their pursuit of economic value while ignoring other significant public values [28,37,38].

*2.3. Public Value Foundation*

Gross Domestic Product (GDP) was invented during the 1930s to 1940s amid the Great Depression and World War II. It has been one of the most prominent metrics for measuring social progress worldwide. Ever since, GDP and economic value have been recognized as the most critical public values, stimulating post-war reconstruction and economic prosperity. Boosting the economy became the most fundamental policy goal for nearly all governments, and economic growth measured by GDP was considered basically equivalent to social progress [39]. However, since the 1960s, GDP as a proxy for social progress has aroused widespread criticism. Not all social aspects and human behaviors can be monetarized as a reference to market price, as GDP is incapable of accounting for other public values excluding the economic values, such as social justice, individual mental health, and non-marketed natural and environmental resources. The economic way of thinking has gained a reputation for knowing the price of everything but the value of nothing [40]. As many invaluable items are missing and materialized by GDP and national income accounting, we appear more affluent than we really are, and we are losing valuable resources [21].

In response to this, the Social Indicators Movement in the 1960s [6], the Beyond GDP Movement in 2007 [4] and the United Nations Sustainable Development Goals (SDGs) in 2015 [8] have attempted to redefine social progress and create replaceable and more comprehensive indicators. For instance, the Human Development Index (HDI), Environmental Performance Index (EPI), Ecological Footprint (EF), and Happy Planet Index (HPI) were invented and compiled, as were the countries that can be measured and ranked by these different indicators. Public values such as equity, education, health, well-being, and ecological sustainability have been considered and encompassed in these indicators [41].

Most indicators have explicit or implicit value foundations. The Human Development Index (HDI), designed by the United Nations Development Programme (UNDP) in 1990, was one of the most prominent candidates to replace GDP. It covers three social domains and the values of income, health, and education with the computational formula

$$\text{HDI} = (\text{I}_{\text{Health}} \times \text{I}_{\text{Education}} \times \text{I}_{\text{Income}})^{1/3}$$

Although the HDI is more comprehensive than GDP, it has also aroused many critiques, many of which focused on its insufficient measurement of human progress and its high correlation with GDP. For instance, some have argued that the HDI depicts an oversimplified view of social development via a few indicators with poor data quality [42]. Additionally, the three components of HDI are strongly correlated with each other, and thus HDI does not reveal anything new beyond GDP or life expectancy [43,44]. Moreover, Sagar and Najam [45] claimed that HDI presents a distorted picture of the world, similar to GDP, and ignores the environmental dimensions of development. To modify these defects,

UNDP developed the Inequality-adjusted Human Development Index (IHDI) in 2015 and incorporated inequality as an adjustment. Further, following the idea of nature-based human development, the Planetary Pressures-adjusted Human Development Index (PHDI) was proposed [12]. However, as they do not clarify the role of public value as a foundation of creating indicators, these endeavors are insufficient since the relationships between various public values are complex and unclear.

Ecology and the environment are increasingly important public values. To measure these values, the Environmental Performance Index (EPI) was invented by the Yale Center for Environmental Law & Policy (YCELP) in 2006 as a global metric to compare and rank the performance of different countries in terms of ecological sustainability. The EPI aggregates 32 indicators across 11 categories, including air quality, sanitation and drinking water, heavy metals, waste management, biodiversity and habitat, ecosystem services, fisheries, climate change, pollution emissions, water resources and agriculture, etc. [13]. Despite no direct economic-based sub-indicators being employed in the EPI, it also has a strong correlation with GDP per capita, revealing the close relationship between GDP and the environment but reducing the novelty of the EPI [46–49].

Another indicator that measures environmental value is the Ecological Footprint (EF), which was designed by Mathis Wackernagel and William Rees in the 1990s. It measures the human demand for biocapacity, e.g., how many hectares of land are consumed per person or population, without requiring it to be transformed to monetary units [50]. As a widely recognized sustainability metric, EF offers an integrated and multiscale path to track human consumption and overshoot of natural resources and their effects on the ecosystem and biodiversity [51]. According to Lin et al. [52], despite the increase in global biocapacity due to the improved management techniques and increased agricultural yields since 1961, the global EF continues to increase at a faster pace than biocapacity. Particularly, rich countries are living parasitically on the planet because their rapid economic growth is established on the overconsumption of non-renewable resources, massive $CO_2$ emissions, and stealing part of the share from developing countries [50,53].

Combining the public values of happiness and ecology, the Happy Planet Index (HPI) was developed by the New Economic Foundation [54] and measures comparable sustainable well-being on a global scale. It was defined as the national Inequality-adjusted Happy Life Years achieved per unit of natural resource use measured by the Ecological Footprint using the equation

$$\text{HPI} \approx \frac{(\text{Life Expectancy} \times \text{Experienced Wellbeing}) \times \text{ Inequality of Outcomes}}{\text{Ecological Footprint}}$$

The HPI creatively introduced several distinctive sub-indicators, such as the Ecological Footprint, and aggregated them in a comparative well-structured format that was free of the dominance of GDP [55]. Hence, its ranking of country performance is much different from most other indicators. Some Latin American countries such as Costa Rica and Mexico and countries in Southeast Asia such as Vietnam and Thailand lead in the rankings and surpassed most developed countries. This result verifies that when we change the indicators, which are based upon different values, the country rankings as well as perceptions and evaluations will change correspondingly.

### 3. Methodology and Data

*3.1. Data Source*

In addition to GDP per capita (GDPP), 12 alternative indicators were selected based on three identified groups, and were global-scale comparable indicators recognized worldwide and considered reliable, were developed via academic endeavors, and were acquired from the official sites or reports (Table 1). Specifically, apart from GDPP as a key reference, 12 other indicators are classified into three groups and measure three types of public values. Group 1 contains indicators related to human well-being; Group 2 refers to those related to ecological sustainability; and Group 3 contains both types of public

values. The 12 alternative indicators that were adopted in our research are influential and representative, other indicators can be included, and the methodology is generally applicable. In addition, the research was a cross-section analysis. Since not all indicators are updated synchronously and since the data does not fluctuate sharply, based on the principle that all selected indicators must have been updated within the last 5 years, the most recently available data were collected. With the exceptions of the HPI and EFP, for which the most recent data were from 2016 and 2017, respectively, data of the others were from 2019–2021. The raw data for selected indicators are provided in the Supplementary Materials. Moreover, due to some of the information being unavailable for some countries, 98 countries were employed in the PCA after data screening.

**Table 1.** Classification, source, and publication year of GDPP as well as the twelve alternative indicators.

| Indicator | Full Name of Indicator | Source | Year |
|---|---|---|---|
| Key Reference | | | |
| 1. GDPP | GDP per capita (current US$) | [56] | 2019 |
| Group 1: Measure Human Wellbeing | | | |
| 1. HDI | Human Development Index | [57] | 2019 |
| 2. IHDI | Inequality-adjusted Human Development Index | [57] | 2019 |
| 3. WHI | The World Happiness Index | [58] | 2021 |
| 4. WGI | The Worldwide Governance Indicators | [59] | 2019 |
| Group 2: Measure Ecological Sustainability | | | |
| 1. PHDI | Planetary Pressure-adjusted Human Development Index | [60] | 2019 |
| 2. EPI | Environmental Performance Index | [61] | 2020 |
| 3. EFP | Ecological Footprint per capita | [62] | 2017 |
| 4. WETI | World Energy Trilemma Index | [63] | 2021 |
| Group 3: Measure both Human Wellbeing and Ecological Sustainability | | | |
| 1. SDI | Sustainable Development Index | [64] | 2020 |
| 2. LPI | The Legatum Prosperity Index | [65] | 2020 |
| 3. SPI | Social Progress Index | [66] | 2020 |
| 4. HPI | Happy Planet Index | [14] | 2016 |

Note: The six sub-indicators of WGI have performance scores ranging from −2.5 to 2.5 without aggregation as a single number [67]. For the convenience of the statistical calculation and analysis, WGI has been treated as a compound index, and the final score is calculated from the arithmetic mean value of scores from its six components.

### 3.2. Data Normalization

We applied a PCA to investigate the relationships between GDPP and the 12 beyond GDP indicators. PCA can deal with double-counted and overlapping information in two or more variables [68] and is a popular tool for constructing composite indices that contain many highly correlated variables. In addition to the primary function of dimensional reduction, it may reveal intrinsic information and correlations among multi-dimensional variables [69]. Hence, PCA is a powerful technique for investigating the complex relationships among the 13 indicators.

An essential step before proceeding with a PCA is to ensure that all variables are measured by the same unit. It requires that the scores of the chosen 13 indicators be normalized into dimensionless and comparable numbers. There are several major normalization methods, including the Min–Max Normalization, Z–Score Normalization, and Decimal Scaling Normalization. In our study, Min–Max Normalization was utilized, as it offers a linear transformation of the original data between 0 and 1. We also used other methods of normalization, and the PCA results were similar.

After data normalization, the final step is to test whether the normalized data are suitable for PCA via the Kaiser–Meyer–Olkin (KMO) test and Bartlett's test of sphericity. Kaiser [70] recommended a minimum KMO value of 0.5, and the current consensus is that a value between 0.5 and 0.7 is considered mediocre, a value between 0.7 and 0.8 is good, and a value between 0.8 and 0.9 is considered great. If the KMO value exceeds 0.9, then the

data are superb for PCA [71]. Additionally, Bartlett's test of sphericity suggests whether the chosen variables are correlated enough to conduct a PCA. Bartlett [72] asserted that the significance should be less than the value of 0.05 ($p < 0.05$) for the PCA to be appropriate. For our chosen data, the KMO value is 0.89, as shown in Table 2, and the *p*-value calculated from Bartlett's test of sphericity is extremely small, that is close to 0, indicating that our indicators and data passed the adequacy tests and that we could proceed with the PCA.

**Table 2.** KMO test and Bartlett's test of sphericity. *Source*: author's calculations.

| Kaiser-Meyer-Olkin (KMO) Test (Overall MSA) | | 0.89 |
|---|---|---|
| **Bartlett's Test of Sphericity** | Chisq | 2572.144 |
| | *p*-value | 0.000 |
| | Df | 78 |

### 3.3. Correlation Matrix

The data satisfy the KMO test and Bartlett's test of sphericity, and the correlations of the normalized data are illustrated in the correlation matrix via the R packages of "GGally" [73] and "ggplot2" [74]. In Figure 1, scatterplots of each pair of numeric variables are shown on the left, Pearson correlations are displayed on the right, while the variable distributions are shown diagonally. Specifically, with the exception of the HPI, most of the other indicators are strongly mutually correlated positively or negatively. The first row of the matrix represents the Pearson correlations between GDPP and other indicators. Apart from PHDI (0.45) and HPI (0.02), the others have high correlations with GDPP (i.e., they exceed 0.60). By contrast, as demonstrated in the last column, the correlations between the HPI and the other indicators are relatively low (i.e., they are less than 0.50), and dots in the scatterplots (last row) are distributed dispersedly (marked by the red box), indicating the weak correlations between the HPI and the other indicators. Remarkably, GDPP and HPI are basically irrelevant, as their Pearson correlation coefficient is merely 0.02, suggesting that HPI is beyond GDPP and measures other different public values. Additionally, unlike other indicators with positive correlations, the EFP negatively correlated with other indicators excluding HPI, and the dots are distributed in the opposite direction from all other scatterplots (marked by the blue box). The Pearson correlation coefficient between GDPP and EFP is −0.80, indicating EFP's apparent conflict with GDPP and economic values.

### 3.4. Principal Components

To launch the PCA, we employed the "FactoMineR" [75], "factoextra" [76], and "corrplot" [77] R packages. Since 13 indicators were used as variables, the PCA can generate 13 components; however, only those with a high eigenvalue, generally higher than 1, are selected as principal components (PCs), and the others are not principal and can be ignored. In our study, the eigenvalues of the first two PCs exceed 1 and satisfy this criterion, and they explain 77.88% and 11.31% of the information in the 13 indicators, respectively (Table 3). Specifically, as Table 3 presents, the first PC explains 77.88% of the information in all indicators. In addition to the HPI, correlations between the first PC and the other 12 indicators are very strong (higher than 0.80), such as the HDI (0.98), LPI (0.98), EPI (0.95) and GDPP (0.81). As these indicators have much in common regarding public values about the economy (in Figure 1, their correlation coefficients with GDPP are generally higher than 0.60), and then the first PC can be roughly defined as the economic well-being values. In our study, $R^2$ of GDPP with the first PC is 65.61% ($0.81^2$), and because the first PC explains the 77.88% of the total variance, then GDPP interprets 51.10% (65.61% × 77.88%) of the information in the total 13 indicators. This is the reason why most indicators in our research are not yet beyond GDPP: although most of them are more comprehensive than GDPP due to their higher correlations with the first PC, they measure values that are similar to those that are measured by GDPP. Due to the orthogonality of the PCs, the second PC, which explains 11.31% of the information of other public values that are irrelevant to economic well-being values, is mainly determined by the HPI and EFP, which have correlation values

of 0.89 and 0.52 with the second PC, respectively. Since they mainly measure well-being and ecological sustainability, the second PC can be defined as the ecological well-being values. The remaining PCs contain and measure other public values, but they are too weak in these 13 indicators, explaining only 10.81% of the information as a whole, which is very low when compared to the economic well-being values and ecological well-being values.

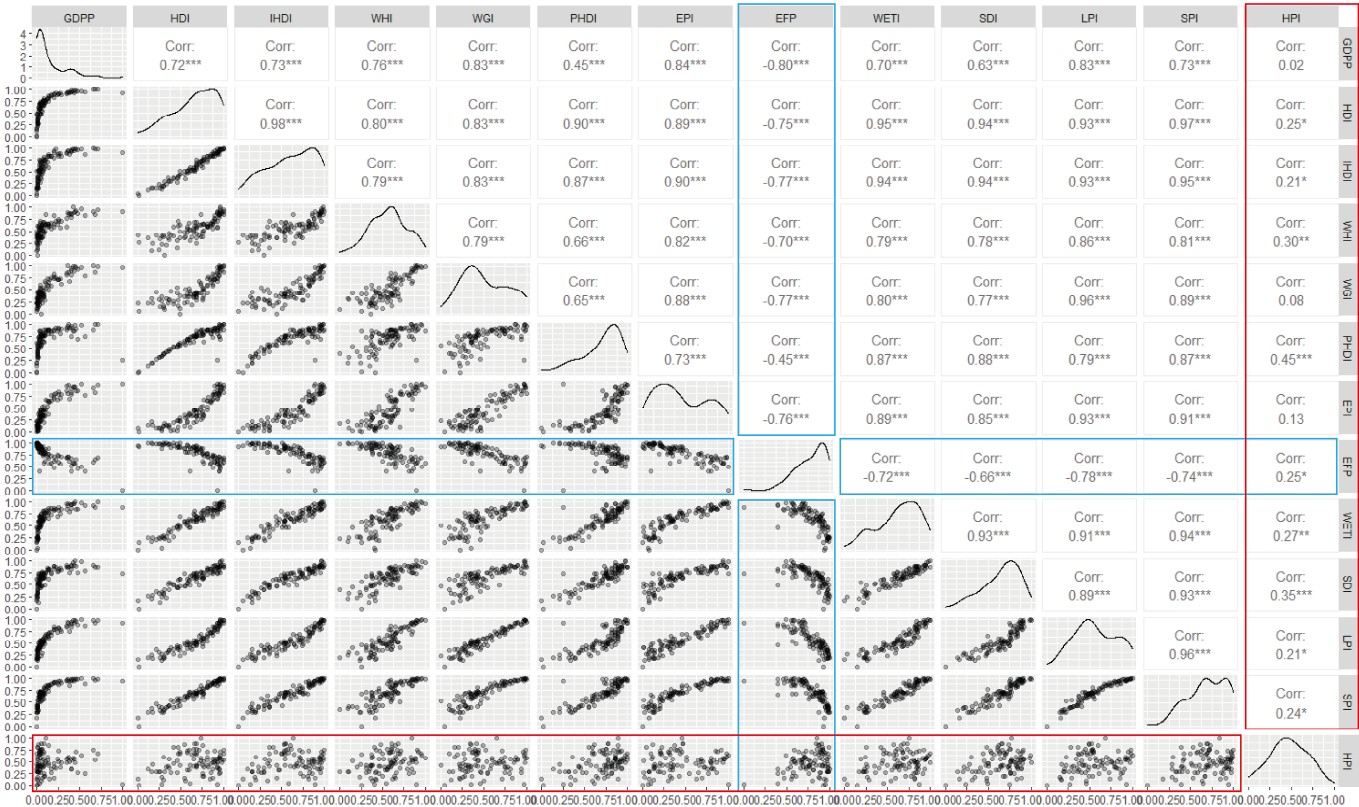

**Figure 1.** Correlation matrix of GDPP and twelve "Beyond GDP" indicators. Note: (1) "*" represents *p* < 0.05, "**" represents *p* < 0.01, "***" represents *p* < 0.001. (2) Red box represents the correlation coefficients and scatterplots between HPI and other indicators, while blue box represents the correlation coefficients and scatterplots between EFP and other indicators.

The correlation circle in Figure 2 displays the indicators' relationships in 13-dimensional space, which are projected in the coordinate system that is constituted by the first two PCs. Strongly correlated indicators closely converge with smaller angles, while weakly and negatively correlated ones are positioned nearly perpendicular or opposite to each other. The length of the arrow represents the information of the indicator kept in the PC1-PC2 dimension, and an indicator closer to the circle's circumference indicates that it occupies a higher proportion or is more represented in this dimension [78]. As shown in Figure 2, all of indicators are close to the circumference of the circle, indicating that they were fully represented by the first two PCs. This means that most information has been preserved in this dimension.

It can be seen that with the exception of the HPI and EFP, other indicators are positioned closely and have small angles around the x-axis (first PC), revealing their high similarities and correlations with GDPP and the economic well-being values. By contrast, HPI is positioned away from other indicators and close to the y-axis (forming the second PC), especially perpendicular to the GDPP, indicating its extremely weak correlation with the GDPP and the economic well-being values. Additionally, the EFP is strongly negatively correlated with GDPP and positioned nearly opposite to it, reflecting opposing values of GDPP. These findings discover that only HPI and EFP can provide novel perspectives

and public values that are very different from GDPP and economic well-being values in defining and measuring social progress.

**Table 3.** Principal component loadings.

| Indicator | Principal Component | |
|---|---|---|
| | PC1 | PC2 |
| GDPP | *0.81* | −0.36 |
| HDI | *0.98* | 0.08 |
| IHDI | *0.97* | 0.04 |
| WHI | *0.87* | 0.01 |
| WGI | *0.91* | −0.20 |
| PHDI | *0.84* | 0.42 |
| EPI | *0.95* | −0.11 |
| EFP | *−0.80* | *0.52* |
| WETI | *0.95* | 0.11 |
| SDI | *0.94* | 0.20 |
| LPI | *0.98* | −0.05 |
| SPI | *0.98* | 0.05 |
| HPI | 0.23 | *0.89* |
| Eigenvalues | 10.12 | 1.47 |
| % Variance explained | 77.88 | 11.31 |
| % Cumulative variance | 77.88 | 89.19 |

Note: Figures in italics indicate absolute values greater than 0.50.

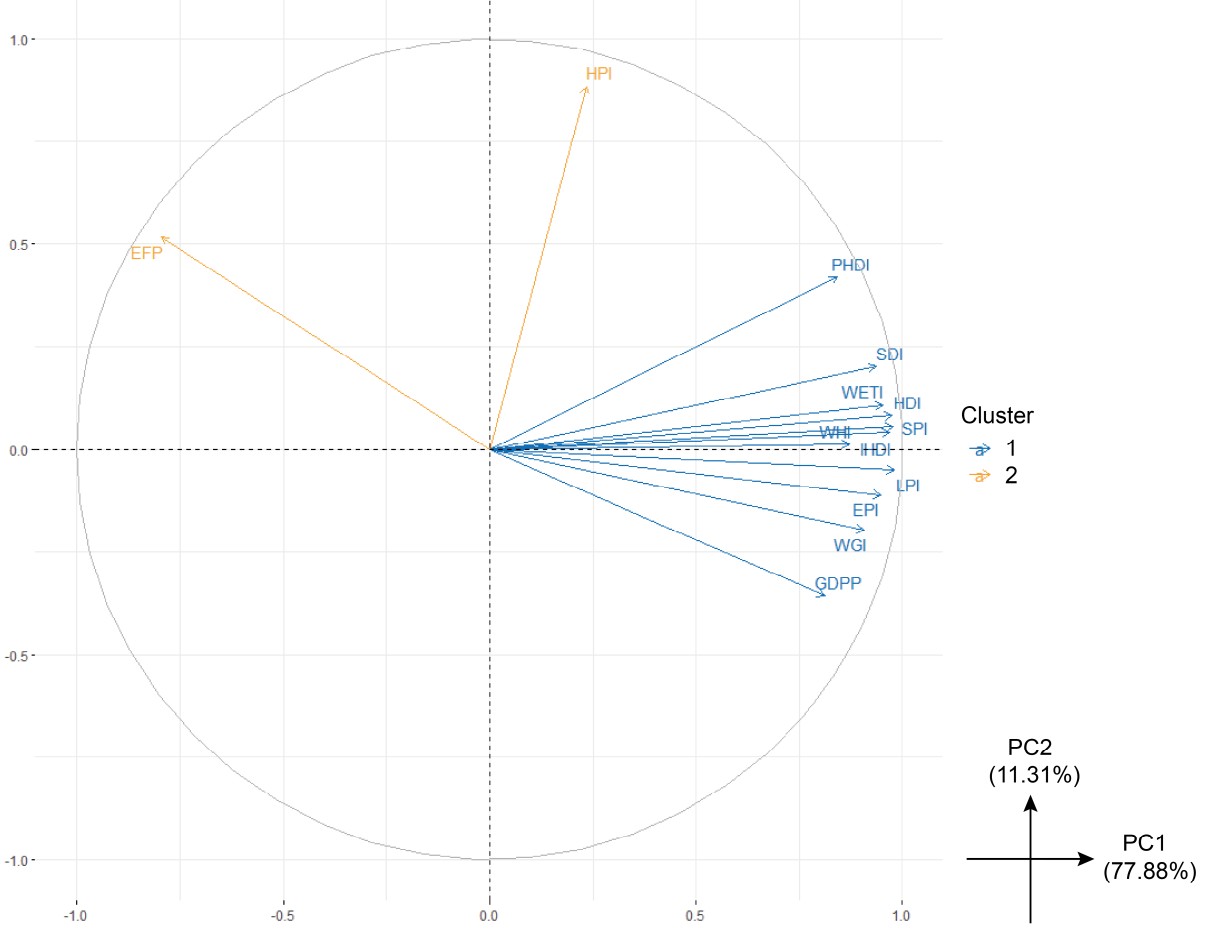

**Figure 2.** Correlation circle.

The biplot in Figure 3 displays the positions of countries on the plane constituted by the first two PCs. A country on the same side of a given indicator has a high value for this indicator and vice versa [78]. As mentioned before, the first PC (x-axis) represents economic well-being values, and the second one represents the ecological well-being values (y-axis). The countries located around the first PC (positive x-axis) are mainly developed countries that have a high GDPP, such as Nordic, North American, and Oceanian countries. Meanwhile, countries from South Asia and sub-Saharan Africa, who lead the EFP ranking but have a low GDPP are on the opposite side (negative x-axis). Regarding the second PC (y-axis), countries from Latin America and Southeast Asia such as Costa Rica and Vietnam perform much better than most developed countries.

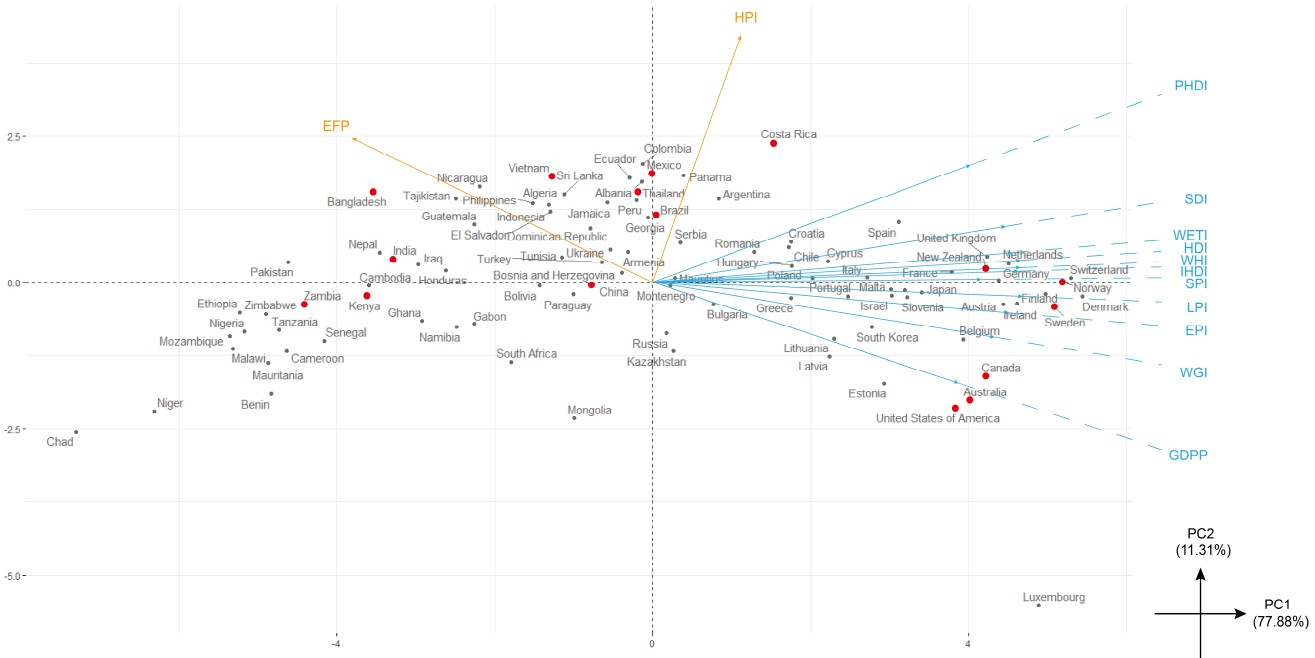

**Figure 3.** Biplot of countries, GDPP, and alternative indicators. Note: Due to some of the information being unavailable for some of the countries, 98 countries were included after data screening.

To better visualize the differences in terms of a country's performance between GDPP and those that have low or negative correlations with it, Figure 4 illustrates the ranking changes of twelve representative countries by HPI, GDPP, and EFP and uses GDPP as a benchmark. Firstly, regarding GDPP and EFP, they are highly negatively correlated, so their rankings are basically upside down, indicating that wealthy and developed countries such as the United States, Australia, and Sweden, which have a higher GDPP, are ranked at the bottom in terms of the EFP, while underdeveloped countries in South Asia and sub-Saharan Africa, such as Bangladesh and Zambia, lead in the EFP rankings. Secondly, regarding GDPP and the HPI, the extremely low correlation between them leads to a shuffling in the rankings. Countries from Latin America and Southeast Asia, such as Costa Rica and Vietnam, take the top positions in terms of the HPI ranking, while the positions of developed countries drop significantly. These dramatic changes in rankings indicate that when we change the value and indicator (for example, from GDPP to EFP or HPI), the best country will change accordingly depending on that value and indicator.

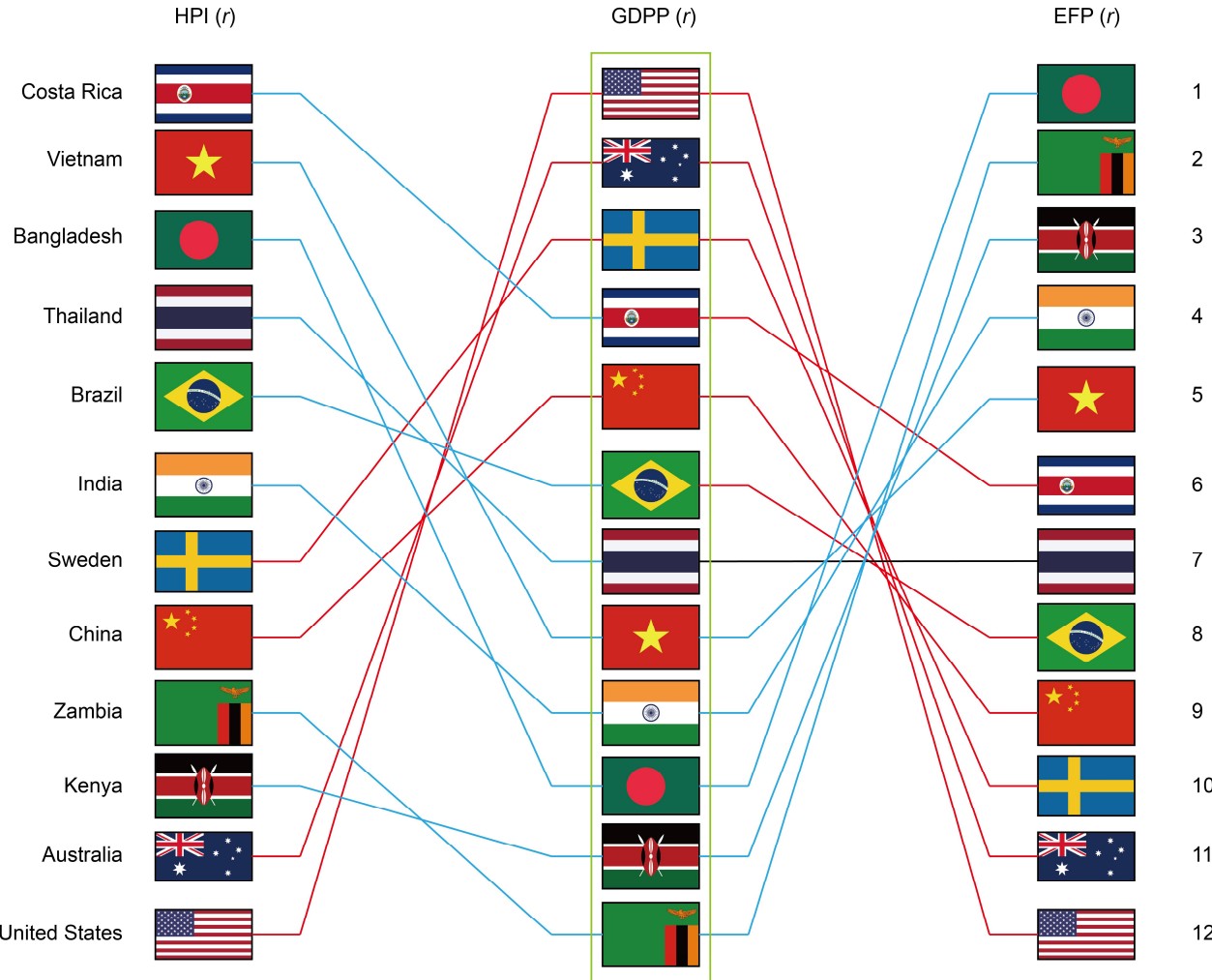

**Figure 4.** Ranking changes of twelve representative countries in terms of HPI, GDPP, and EFP. Note: (1) Twelve representative countries are selected based on their performance in HPI, GDPP, and EFP. Specifically, most countries distributed around the first PC (economic well-being values) are developed countries from North America (United States), Oceania (Australia), and Northern Europe (Sweden). Countries near the second PC (ecological well-being values) are mostly from Latin America (Costa Rica, Brazil) and Southeast Asia (Vietnam, Thailand). Moreover, most countries distributed along the EFP are from South Asia (India, Bangladesh) and sub-Saharan Africa (Zambia, Kenya). Combined with China as a significant case, these 12 countries were ranked from 1 to 12 based on their GDPP and then were reranked in terms of HPI and EFP, with the purpose indicating how countries that rank high in GDPP perform in HPI and EFP. (2) Taking GDPP as a benchmark, a red line represents a drop in ranking, a blue line represents a rise, while a black line represents the same ranking.

## 4. Results and Discussion

### 4.1. Major Role of GDP

The PCA revealed that most of the selected indicators in our study have strong correlations with GDPP, despite their inclusion of well-being and environmental factors. Specifically, although most of indicators in our research are more comprehensive than GDPP due to their higher correlations with the first PC, GDPP can still explain 51.10% of the information in the total 13 indicators, indicating its major role in social progress. This is why most indicators in our research do not go beyond GDPP. The wealthier countries generally rank higher in most indicators, and developed countries generally occupy most of the top 20 places in various indicator rankings, meaning that they are recognized as

countries leading human progress [79]. Our research discovers that GDPP and economic well-being values play major roles in the social progress, and thus it is difficult for other indicators and public values to move beyond them.

Economic well-being values, which are usually measured by GDP and GDPP, are easy to be quantified and understood by both politicians and the general public. In addition, GDP can be projected for years or even further into the future, enabling policymakers and economic institutes to monitor and judge whether the economy is contracting or expanding, allowing them to take prompt and necessary actions. Particularly, as the PCA discovered, GDPP can explain 65.61% of the information in the economic well-being values (first PC) and 51.10% of the total 13 indicators. Hence, GDP growth is able to drive the progress of other social domains such as education, healthcare, and infrastructure, indicating that it can act as a simple and excellent indicator for measuring the bulk of the social development [21]. Therefore, Beyond GDP indicators actually supplement and enhance GDP but do not replace or move beyond it [80].

However, there are social domains and public values that are inconsistent with economic ones, such as political, cultural, social, and ecological values. Their progress may not be synchronous and coordinated. Developed and developing countries have very different public value concerns that are dependent on different cultural backgrounds and different stages of social development [36]. Regarding developing countries, economic growth is urgent in the early development stages, and other public values, such as economic equity and environmental protection, may be ignored [21]. It is difficult for plural public values to coexist with rapid economic growth during the initial development phases [81]. In contrast, developed countries have already established sufficient economic foundations, enabling them to concern themselves with other public values, such as social justice and ecological sustainability. Even so, when confronting economic stagnation or crises, these countries are still urged to pursue economic growth and employment and place less emphasis on other public values. Social justice and environmental beauty are generally luxury goods and are only considered seriously when the economy is booming [16].

### 4.2. Real Flaws of GDP

Despite the major role of GDP in social progress, the sole and excessive pursuit of GDP is not sustainable. With GDP growth, public values in other social domains may be ignored and destructed, such as inequalities in income and wealth, social injustice, environment degradation, and psychological depression [20]. As discovered by the PCA results, EFP has high negative correlations with most indicators that positively correlate with GDPP, and HPI correlates with GDPP by merely 0.02, as Figure 1 shows. These two indicators have diagnosed some potential issues that are hidden by GDP. Although most developed countries lead in most of the indicators that are positively correlated with GDP and are potentially and unconsciously regarded as the best countries, their rankings in EFP and HPI drop dramatically. This means that, currently, human progress is based on the mass consumption of natural resources and most developed countries have consumed large amounts of resources to propel their economies, which has exerted huge pressures on the planet and brought about adverse influence on the people's subjective well-being. Therefore, indicators with negative or weak correlations with GDPP, such as the EFP and HPI, can offer us novel and enlightening information on social progress.

The flaws of GDP primarily lie in its prominent features: modernization and monistic value. The so-called progress is based upon the modernization of an industrial civilization. Most indicators equate social progress with modernization subconsciously, measuring similar things and values. Although modernization introduces enormous values such as technological advancement and better living standards, it also destroys many other values, deteriorating psychological well-being [82], creating lower-quality interpersonal relationships [83,84], and causing ecologically destructive lifestyles [85]. Because of massive and ubiquitous business advertisements, people indulge in consuming commodities that they

do not really need, deteriorating their well-being and the natural environment. Rational consumption is actually not rational but is instead provoked [86].

Neither the EFP nor HPI can be a complete metric of sustainability or human well-being since human development is multi-dimensional and based upon plural public values, but they can provide novel and enlightening information that is different from the information provided by the GDPP. The EFP can measure biological capability per person and indicate that today's economy is based on the overconsumption of non-renewable natural resources; it is also able to determine that this development pattern is unsustainable [87]. The countries that are ranked highly based on the EFP are primarily in South Asia and sub-Saharan Africa, but this does not mean they have outstanding performance in terms of sustainable development, but rather, they tend to have low economic capability. Under the current industrialization paradigm, these countries are still eager to pursue GDP growth but not while decreasing EFP, and we need to seek a new paradigm for progress where the GDP grows by the lower EFP [88].

Regarding the HPI, although some countries in Latin America and Asia, such as Costa Rica, Mexico, Argentina, Brazil, and Thailand, rank high on the HPI and perform well in terms of the ecological well-being values, they also suffer from some public value defects, such as serious income inequality and the middle-income trap [89,90]. Brazil, for example, is not only one of the countries that records the highest income inequality, but it also does not perform well on institutional quality and national governance capabilities [59,91]. Therefore, although HPI can diagnose potential problems and provide information that GDP cannot, it should be supplementary and should not replace GDP.

*4.3. Social Progress beyond GDP*

The advocates and critics of GDP are reasonable but not comprehensive. The former recognizes the major role of GDP in social progress but ignores other public values during the process of economic growth. In contrast, the critics of GDP recognize its flaws but do not consider its major role in social progress and public values. In our research, GDPP can explain 65.61% of the information in the economic well-being values (first PC) and 51.10% of the information in the total 13 indicators. Hence, GDPP still plays a significant role in improving economic well-being and promoting social development. Governments are responsible for comprehensive social development. The key strategy is to prioritize plural public values in a timely and effective manner as well as to arrange their balance and optimal configuration in different development phases. Policy failures usually result from the improper prioritization of public values, such as sufficient economic growth and insufficient social justice and environment protection [36]. It is essential for governments to form a comprehensive and systematic vision of plural public values as well as to determine the value priority and implement policy transformations.

China could be a good case to demonstrate the dynamic prioritization of public values during its journey towards social progress. Since the reform and opening-up policy in 1978, the country achieved rapid GDP growth of about 11.3% until 2020 over the course of four decades, representing the highest GDP growth in the world and creating tremendous economic well-being values, making China's economy the number two world power. However, this rapid economic growth has also led to the neglect and destruction of public values in other social domains. The economic gap has widened sharply, around 0.4 as measured by the Gini coefficient in 2010 from about 0.2 in the 1980s. The environment has been heavily polluted, as evidenced by the serious air smog in urban areas. Political corruption has become increasingly severe, ranking 78th in 2010 as measured by the Corruption Perception Index compiled by Transparency International. China's political leaders and Communist Party have realized these issues and have advocated for the Five Domains of Comprehensive Development since 2012, representing how the economy, politics, society, culture, and ecology should develop synchronously and in coordination with each other [36].

After ten years of effective transformation and governance, China has achieved prominent success in these five domains and in corresponding public values' creation, including poverty reduction, environmental protection, and anti-corruption campaigns. Its HDI rank climbed from number 100 in 2012 to number 85 in 2020 [12]. The headcount poverty ratio has been reduced by 94% from 1980 to 2015 in rural areas, contributing to more than 70% of the reduction in global poverty over the past four decades [92,93], and absolute poverty was eliminated in 2020, an achievement that was announced by the government. Economic growth plays an overwhelming role in poverty reduction, but governments need to shift policy concerns and take effective measures [94,95]. In the political domain, after an iron-fisted anti-corruption campaign, China's Corruption Perception Index jumped from 78th in 2010 to 66th in 2021. China's social progress exhibits a clear prioritization shift in public values, emphasizing GDP growth in the early stages and then making timely adjustments afterwards. The primary logic for this is that the creation of economic well-being values has strengthened China's economic capability, enabling it to focus on other dimensions of public values and allowing the country to achieve more balanced development.

## 5. Conclusions

The measure of GDP has long been criticized as a proxy for social progress, but its merits and demerits have not been fully understood and argued. Although numerous alternative indicators, such as the 12 chosen in our study, have been introduced by the Beyond GDP Movement to replace or supplement GDP, most of them are not entirely beyond GDP. The PCA performed in this research clearly exhibited the multiple dimensional relationships of GDPP and alternative indicators. Although most of the selected indicators are more comprehensive than GDPP due to their higher correlations with the first PC, GDPP still plays a major role (65.61%) in economic well-being values (first PC) and the total 13 indicators (51.10%). Only the EFP and HPI were able to reveal opposing or different information and can be regarded as beyond GDP. This study has clarified the public value foundation of various indicators, helping us to better understand the complex relationships between different social domains and their public values. Social progress can be defined as the enhancement of various public values in different social domains. Measures of GDP and economic well-being values are dominant but are not the sole indicator or value of comprehensive social progress.

The merit of GDP is that it can easily quantify and measure economic well-being values and enable governments and the general public to grasp the primary aspects of social progress. Based on our research, GDPP plays a major role in the first PC (economic well-being values) and the total 13 indicators, hence, it can drive the progress of most other social domains and corresponding public values. However, this progress is not always synchronous and orderly because different countries have diverse public value concerns since they are at different stages of social development. Developing countries may emphasize more on rapid economic growth and economic well-being because they are in the early phases of development, and other public values such as equity and the environment may be ignored. In contrast, developed countries have already established sufficient economic foundations and may be able to focus more on other public values. However, regardless of whether a country is developing or developed, when facing economic downturns or crises, the economic well-being values and GDP growth will be highlighted, and the priority of other public values may be lowered. The dominance of GDP and economic well-being values is the reason why it is difficult to move beyond GDP.

Despite playing a major role during social development, GDP cannot represent all public values or tell the full story of social progress. The excessive and narrow pursuit of GDP growth may lead to other public values being overlooked or deteriorating. In contrast to indicators such as the HDI, which is highly positively correlated with GDPP, indicators such as the EFP and HPI, which have negative or weak correlations with GDPP, should be considered specifically since they can provide novel perspectives and can measure different values. Developed countries are not always ranked highly according to these new values

and indicators, while developing and relatively poor countries can also be ranked higher if we consider other values and indicators.

Both the advocacy and criticism of GDP are reasonable but not comprehensive. Social progress is based upon various public values and can be measured by different indicators. Governments should form a strategic view to prioritize different public values, introduce effective policies to arrange these values, and pursue sustainable and comprehensive social progress. No single indicator can tell the whole story. The measure of GDP can still indicate major developments in progress, but other indicators and public values cannot be ignored, and those weakly or negatively relevant indicators should be considered more often since they are truly beyond GDP and reveal different information. Only when we pursue and create comprehensive public values will social progress be achieved, and only then can our countries and the world become better.

**Supplementary Materials:** The following supporting information can be downloaded at: https://www.mdpi.com/article/10.3390/su14116430/s1, Table S1: raw data for selected indicators.

**Author Contributions:** Conceptualization, T.C.; data curation, T.C.; formal analysis, T.C.; funding acquisition, B.W.; project administration, B.W.; supervision, B.W.; visualization, T.C.; writing—original draft, T.C.; writing—review and editing, B.W. All authors have read and agreed to the published version of the manuscript.

**Funding:** This work was supported by Huazhong University of Science and Technology Double First-Class Funds for Humanities and Social Sciences.

**Data Availability Statement:** Raw data for selected indicators in this study are available in the Supplementary Materials.

**Conflicts of Interest:** The authors have no potential conflict of interest to declare with respect to the research, authorship, and/or publication of this article.

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
