# Peer review of "Social Progress beyond GDP: A Principal Component Analysis (PCA) of GDP and Twelve Alternative Indicators"

_sustainability, doi:10.3390/su14116430_

Round 1

Reviewer 1 Report

This is an interesting paper on an important topic, the validity of GDP as a proxy for economic well-being from a Chinese perspective.

Having said this, the paper can certainly not be published in its present shape. To start with, the English is understandable, but it must definitely be fixed with professional editing before.

Second, and the graphs are not digestible: resolution too and characters too small. Also, they need to be explained better.

Last but not least, but the flow of the argumentation lacks stringency and precision. I will address a few major points that stand out, but more must be done in repeated rounds of stringency checks.

  1. The motivation of the paper should be highlighted in the beginning. I only find it superficially mentioned in passing in section 3.1. ("As most researches focus on developing new indicators rather than the in-depth re-examination of the correlations among existing indicators, we attempt to explore the relationships between GDP and the twelve alternative indices to examine whether and how they are beyond GDP.)
  2. There is confusion between GDP levels and growth as indicators for levels of social-well-being or social progress.
  3. The discussion of "Beyond GDP" is scattered around the paper and not sufficiently clear. It should be presented in carefully written paragraph close to the beginning of the text.
  4. The discussion of "value" or "public" value is not clear and attention should be devoted to point out where it departs from the economic mainstream. the same applied to "goodness".
  5. The description of the HDI index can be skipped, else it should go more into detail. Also, the statement the HDI does not reveal anything new beyond GDP is too general.
  6. The 12 indicators, their properties and the reasons for their selection should be discussed in more depth.
  7. The pretests before PCA should be discussed in more depth.
  8. A p value of p=0 does not make sense, better p < 0.01 or p < 0.001, whatever is applicable.
  9. Figure 1 is nearly impossible to read and the scatter-plots under the diagonal should be discussed in more depth.
  10. There are 12 indicators, but how many observations and at what frequency and covering which period?
  11. The statement "the first component explains 77.9% of the information of 302 all indicators, converging to GDP per capita, and then can be defined as the economic value factor" does not make sense to me. What is the correlation of the first component to GDP?
  12. The scree plot can be skipped. The information is in the text.
  13. The "correlation circles" and the "biplots" are impossible to decipher, and they must be explained in more detail.
  14. Figure 5 is confusing. I suggest to replace it with rank correlations.
  15. I find section 4.2. not convincing. GDP is not about "modernization and monistic value". It is just value added according to what the convention refers to.
  16. The arguments of section 5. should refer more to the empirical findings and relate the to the "Beyond GDP" literature.

I would recommend a complete overhaul of the manuscript, followed by professional language editing before a resubmission to this journal or a submission elsewhere.

Reviewer 2 Report

The subject is definitely interesting and within the scope of the Journal.  However, as it stands, much effort needs to be done because the English language is very poor and I have issues with some basic elements around the theses of the authors.

  1. English language

I must admit that I started to make changes myself and ended up re-writing the paper, at which point I stopped.  I honestly believe a thorough external english language editing is compulsory.

2.  GDP and social progress

On page 3, the claim is that GDP measures social development and progress (line 111 and line 134 for example).  Your criticisms are completely right if you take the hypothesis that GDP does measure social development.  As far as I am concerned, the role of GDP is NOT to measure social development but to measure the economic health of a country.  If you take that alternative view, some criticisms are therefore milder.  I do not mean to say that GDP is perfect, far from it, and you raise important issues (equity, ecological problems, sustainability, ...).  These issues are enough to undertake the PCA analysis but I think saying that GDP is synonymous to social progress and development goes too far as it is not what most economists are saying.  I really think this should be discussed in the paper

3.  Other remarks

a) the Figures are very interesting but when printed, most of them are hardly readable

b) The definition of Figure 2 is not correct - scree plot??

c) saying that HPI and EF are very different than most indicators and among others than the GDP related indicators is obviously correct since they do not include GDP in their computations.  It feels like opening an open door...  But that being said, your results expressed in Figure 5 are important and interesting and should be explained in light of the dichotomy GDP rich countries on one side and HPI-EF and poorer countries or countries that are environmentally conscious (like Costa Rica) on the other side. 

Reviewer 3 Report

The topic is interesting, the paper is well structured, however I have few recommendations.

The abstract should indicate country sample and analysis period.

The Methodology and Data section should be improved, considering these aspects:

  1. country sample and analysis period.
  2. what kind of correlation is considered – linear (Pearson coefficient) or rank (Spearman)? Figure 1 indicates that relationship is not linear in some cases.
  3. why correlations are calculated with various indices and GDP in different year? As example, GDP in 2019 while Happy Planet Index is for 2016? Maybe that is the reason of weak correlation?
  4. various indices represent real situation and comparison among countries; however, GDP is measured in current US dollars and is not suitable for real economic situation’s comparison across countries. Why nominal (current prices) GDP is chosen instead of GDP per capita, PPP (current international $)?
  5. on what basis were selected 12 representative countries (Figure 5)?

It is worth to discuss the negative correlation between GDP and Ecological footprint, as these results conflict with other studies, indicating positive relationship.